# Hospital Admission Patterns in Adult Patients with Community-Acquired Pneumonia Who Received Ceftriaxone and a Macrolide by Disease Severity across United States Hospitals

**DOI:** 10.3390/antibiotics9090577

**Published:** 2020-09-04

**Authors:** Thomas P. Lodise, Hoa Van Le, Kenneth LaPensee

**Affiliations:** 1Albany College of Pharmacy and Health Sciences 106 New Scotland Ave, Albany, NY 12208-3492, USA; 2PAREXEL International 2520 Meridian Parkway, Suite 200, Durham, NC 27713, USA; hoa.vanle@gmail.com; 3Paratek Pharmaceuticals, Inc. 1000 1st Avenue, King of Prussia, PA 19406, USA; kennethlapensee@comcast.net

**Keywords:** community-acquired bacterial pneumonia, community-acquired pneumonia, pneumonia severity index, length of stay, mortality, patient readmission

## Abstract

(1) Objective: There are limited data regarding community-acquired pneumonia (CAP) admissions patterns in US hospitals. Current expert CAP guidelines advocate for outpatient treatment or an abbreviated hospital stay for CAP patients in pneumonia severity index (PSI) risk classes I–III (low risk); however, the extent of compliance with this recommendation is unclear. This study sought to estimate the proportion of admissions among CAP patients who received ceftriaxone and macrolide therapy, one of the most commonly prescribed guideline-concordant CAP regimens, by PSI risk class and Charlson comorbidity index (CCI) score. (2) Methods: A retrospective cross-sectional study of patients in the Vizient^®^ (MedAssets, Irving, Texas) database between 2012 and 2015 was performed. Patients were included if they were aged ≥ 18 years, had a primary diagnosis for CAP, and received ceftriaxone and a macrolide on hospital day 1 or 2. Baseline demographics and admitting diagnoses were used to calculate the PSI score. Patients in the final study population were grouped into categories by their PSI risk class and CCI score. Hospital length of stay, 30-day mortality rates, and 30-day CAP-related readmissions were calculated across resulting PSI–CCI strata. (3) Results: Overall, 32,917 patients met the study criteria. Approximately 70% patients were in PSI risk classes I–III and length of stay ranged between 4.9 and 6.2 days, based on CCI score. The 30-day mortality rate was <0.5% and <1.4% in patients with PSI risk classes I and II, respectively. (4) Conclusions: Over two-thirds of hospitalized patients with CAP who received ceftriaxone and a macrolide were in PSI risk classes I–III. Although the findings should be interpreted with caution, they suggest that there is a potential opportunity to improve the efficiency of healthcare delivery for CAP patients by shifting inpatient care to the outpatient setting in appropriate patients.

## 1. Introduction

Despite advances in the care of patients with community-acquired pneumonia (CAP), the associated morbidity, mortality, and costs are still considerable [1,2,3,4]. Pneumonia was the eighth most costly condition for US hospitals in 2013, resulting in direct annual medical costs of over $17 billion [4,5]. Although a variety of factors—including pharmacy, outpatient clinic, and physician office visits—contribute to the direct annual cost of CAP, hospital inpatient care accounts for approximately 80–95%, mostly due to room and board [5]. Since the costs associated with outpatient management of CAP are considerably lower than inpatient care, the official clinical practice guidelines for the diagnosis and treatment of adults with CAP by the Infectious Diseases Society of America/American Thoracic Society (IDSA/ATS) recommend that clinicians use site of care severity of illness indicators and prognostic models to assist in identifying patients with CAP who may be candidates for outpatient treatment [6,7]. The expert guidelines preferably recommend the pneumonia severity index (PSI) [8,9] to assess the need for hospitalization in adults with CAP [7]. The PSI stratifies patients into five classes and recommends that patients in risk classes I and II are treated as outpatients, patients in risk class III are treated in an observation unit or with a short hospitalization (many of these patients may be candidates for outpatient treatment), and patients in risk class IV or V are treated as inpatients [6,7,8,9]. Despite this level I evidence guideline recommendation, there appears to be significant variation in admission rates among hospitals and individual clinicians, indicating differences in site of care decision making across key stakeholders [3,10]. However, there have been few published assessments of admissions rates by PSI risk class among patients with CAP across US hospitals in recent years [11,12].

This study sought to assess hospital admission patterns of adult, hospitalized patients with suspected or documented community-acquired bacterial pneumonia (CABP), a subset of CAP, in a large hospital database. The primary goal was to estimate the proportion of hospital CABP admissions that occur among low-risk patients (PSI risk classes I–III) where outpatient treatment or a brief inpatient stay is advocated. The analyses were limited to those who received empiric ceftriaxone + azithromycin, as use of this combination is highly specific to a CABP diagnosis and it is the most frequently used empiric antibiotic regimen for hospitalized patients with suspected or documented CABP [13].

## 2. Methods

A retrospective study of adult, hospitalized patients in the Vizient^®^ (formerly known as MedAssets Health System) database between July 2012 and June 2015 was conducted. Vizient^®^ (MedAssets, Irving, TX, USA) is a hospital database that incorporates the elements of the CMS Uniform Billing (UB-04) hospital discharge abstract. The MedAssets Health System data include inpatient and hospital-based outpatient data for more than 400 hospitals across 42 states (South, 59%; West, 17%; Midwest, 13%; Northeast, 12%) in the US. The database includes hospitals with large and small facilities in both urban (87%) and rural (13%) locations. Approximately 98% of providers submit both inpatient and outpatient data. Data are updated twice a month with an approximate 30- to 45-day lag from end of month. Within the database, episodes of care contain information on patient encounters grouped by an outpatient visit or inpatient stay with all the associated Current Procedural Terminology procedure codes and International Classification of Diseases, 9th revision, clinical modification (ICD-9-CM) diagnosis codes. Within each episode, ICD-9-CM procedure codes and standardized charge identifications provide details regarding the procedures, medications, services, supplies, and equipment received each day while providing patient care. No clinical laboratory values were available in the database.

Patients were included if they were ≥18 years old, had a primary admission ICD-9-CM diagnosis of CAP (Appendix A)**,** received IV ceftriaxone with a macrolide on hospital day 1 or 2, and had continuous enrolment in the Vizient^®^ database for ≥12 months prior to the index CAP admission. For patients with ≥1 CAP admissions, the first admission was included. As described previously, patients were excluded from the CAP cohort if they had the following diagnoses during the pre-index period: “amputation for vascular insufficiency; aspiration or post-obstructive pneumonia; contiguous cellulitis with a diabetic foot ulcer, psoriasis, or eczema; cystic fibrosis; ecthyma gangrenosum; endocarditis or hospital-acquired pneumonia; infected decubitus ulcers; meningitis; myonecrosis; necrotizing fasciitis; osteomyelitis; perirectal abscess or perineal infection; pregnancy; progressive gangrene; septic arthritis; pyogenic arthritis; infective arthritis; septic shock and severe sepsis; or tuberculosis” [14]. Patients were also excluded if they resided in a nursing home or long-term care facility prior to index CAP admission or if they were transferred from another hospital. These individuals were excluded from the study to create a more homogenous CAP cohort and to exclude potential CAP patients at baseline that would have a likely reason(s) for admission other than their CAP.

Data elements collected in this study are listed in Table 1.

Demographic elements captured in the Vizient^®^ dataset included age at index date, gender, 12-month baseline comorbidities, PSI score [8,9] at admission, and 12-month baseline CCI score [15]. Additional variables included 12-month pre index hospital-based inpatient visits, 12-month pre-index ED visits, and 12-month pre-index hospital-based outpatient visits. The calculation of the PSI score in this study relied on demographic information and ICD-9-CM codes from the day of admission, as laboratory values and physical exam findings were not available (Appendix A). The ICD-9-CM codes for PSI score criteria and adaptations are listed in Appendix A. Of note, patients with neoplasms of the skin were scored as having cancer when calculating their PSI score. However, neoplasms of the skin were not included as one of the cancer types in the original PSI scoring system.

### Data Analyses

Demographics, baseline characteristics, and outcomes were reported as counts and percentages for categorical variables and mean (standard deviation) of central tendency for continuous variables. Patients were grouped into categories by their PSI risk class and CCI score. Hospital length of stay, 30-day mortality rates, and 30-day CAP-related readmissions were calculated across resulting PSI risk class–CCI strata. SAS software version 9.3 (SAS Institute, Cary, NC, USA) was used to complete the analyses.

## 3. Results

The database included 284,549 patients with CABP, which represented 132,473 inpatient admissions. During the study period, 32,917 of the 132,473 hospitalized patients with CABP received ceftriaxone and a macrolide. Approximately 52% of the population was female, and most patients were over 65 years of age (65%) (Table 1). Overall, approximately two-thirds of patients were in PSI risk classes I–II (35%) or PSI risk class III (34%). As shown in Table 2, most patients in PSI risk classes II and III had a CCI score of 0 or 1.

The mean (SD) hospital length of stay (LOS) for patients in PSI risk classes I–II or PSI risk class III ranged between 4.9 (2.8) and 6.2 (3.8) days, depending on the CCI score (Table 2). The 30-day mortality rates for patients in PSI risk classes I–II or PSI risk class III ranged between 0.2% and 1.4%, depending on the CCI score (Appendix A). The rates of hospital readmission for patients in PSI risk classes I–II or PSI risk class III ranged between 3.8% and 9.5%, respectively, depending on the CCI score (Appendix A).

## 4. Discussion

In this analysis of data from more than 400 US hospitals, we found that potentially two-thirds of patients who were hospitalized with CABP and treated with ceftriaxone and a macrolide were in PSI risk classes I–III. According to IDSA/ATS CAP treatment guidelines, most of these patients potentially could have received care in the outpatient setting (PSI risk classes I–II), or they could have been treated in an observation unit or with a short hospitalization with subsequent treatment in the outpatient setting (PSI risk class III) [6,7]. However, the mean hospital LOS ranged between 5 and 6 days for patients in PSI risk classes I–III, which is much longer than the short hospitalization stay recommended in the IDSA/ATS CAP treatment guidelines. Consistent with older data [3,9,10], the findings suggest that inconsistent criteria may be applied for initial decisions about appropriate sites of care and there is a potential opportunity to shift site of care for many hospitalized patients with CABP. Although the results should be interpreted cautiously due to the nature of the analyses, the findings can potentially serve as the basis for quality assurance initiatives in hospitals seeking to improve the efficiency of healthcare delivery for CABP patients. One way to potentially shift the care of CABP to the outpatient setting is by using antibiotics with equivalent IV and oral formulations. Switching patients with CABP from IV to oral antibiotics has been shown to divert unnecessary hospitalizations, reduce hospital LOS, and shorten the duration of IV antibiotic treatment compared with usual care, without compromising patient outcomes or increasing hospital admission rates [16,17,18].

There are several factors to consider when interpreting the findings of this study. Laboratory or other electronic medical record information was not available in the MedAssets database. We therefore had to use ICD-9-CM diagnosis codes from the day of admission to derive the PSI score for each patient. It is quite possible that not all clinical conditions defined by laboratory values and physical exam findings in the PSI scoring system were coded. However, the major drivers of PSI scores are age, sex, and comorbid conditions, and these were adequately captured in the MedAssets database. Furthermore, the mortality rates among patients in PSI risk classes I–III were found to be very low in this study and consistent with the original study by Fine et al. [8], suggesting that our algorithm for calculating the PSI score was fairly accurate in estimating the mortality risk. As a further safeguard, we stratified each PSI risk class by their CCI score and most patients in PSI risk classes I-III had CCI scores ≤ 1. This suggests that the lower PSI risk class patients were not being admitted due to underlying comorbidities. In addition, 30-day CABP-related readmission rates were also found to be low, providing additional support for use of the algorithm for calculating the PSI score in this study.

It is also important to note that the initial severity of a patient’s CAP is not the only predictor of hospital admission and LOS. Community-acquired pneumonia may worsen other comorbidities such as chronic heart failure, COPD, and renal failure, which may increase hospitalization time. The influence of this aspect of CAP hospitalization was not evaluated in this study and should be investigated further. We were also not able to fully elucidate the clinical and non-clinical (e.g., socioeconomic) reasons for hospital admission among these patients. We also could not determine whether patients would have had similar or better outcomes with an alternative site of care. The intent of this study was simply to highlight the proportion of adult, hospitalized CAP patients that are considered “low risk” and could potentially be managed in the outpatient setting. Given the annual burden of CAP in the US (i.e., 1 million admissions per year) [4], even a conservative application of our findings suggests that there is a tremendous opportunity to improve the efficiency of healthcare delivery for CABP patients by shifting inpatient care to the outpatient setting in appropriate patients.

In summary, over two-thirds of hospitalized CABP patients who received ceftriaxone IV and a macrolide in this analysis were estimated to be in PSI risk classes I–III. On average, hospital LOS was 5–6 days for patients with CAP who were in PSI risk classes I–III. According to expert CAP treatment guidelines [6,7], many of these patients could have received their care in the outpatient setting or in an observation unit because of the low mortality risks associated with these PSI risk classes. Although the results should be interpreted with extreme caution due to the inherent limitations associated with the analyses, the findings suggest that there is an opportunity improve the efficiency of healthcare delivery by shifting the initial site of care from the inpatient to the outpatient setting in appropriate “low-risk” CABP patients. Similar to other healthcare database studies, further investigations in the clinical arena are needed. It will be important to determine whether the admission patterns by PSI risk class observed in this study are consistent with future studies of a similar nature that use laboratory data to calculate the PSI score.

## Figures and Tables

**Table 1 antibiotics-09-00577-t001:** Characteristics of study population (*n* = 32,917).

Characteristics	Patients, *n* (%)
Female	17,028 (51.7)
Age ≥65 years	21,484 (65.3)
PSI score category	
≤70 (risk class I and II)	11,589 (35.2)
71–90 (risk class III)	11,100 (33.7)
91–130 (risk class IV)	9349 (28.4)
>130 (risk class V)	879 (2.7)
CCI score	
0	7303 (22.2)
1	9056 (27.5)
2	6295 (19.1)
3+	10,263 (31.2)
Prior CAP	1792 (5.4)
Coronary heart disease	14,725 (44.7)
Diabetes	9454 (28.7)
Chronic pulmonary disease	6983 (21.2)
Acute respiratory failure	5782 (17.6)
Immunocompromising conditions	4865 (14.8)
Surgery without implant	5613 (17.1)
Congestive heart failure	4825 (14.7)
Acute renal failure	4780 (14.5)
Bronchiectasis or severe COPD	4233 (12.9)
Cancer	2942 (8.9)
Dementia	2550 (7.7)
12-month, pre-index, hospital-based inpatient visit	1556 (4.7)
12-month, pre-index, ED visit	1234 (3.7)
12-month, pre-index, hospital-based outpatient visit	488 (1.5)
Severe renal disease	1872 (5.7)

CAP, community-acquired pneumonia; CCI, Charlson comorbidity index; COPD, chronic obstructive pulmonary disease; ED, emergency department; PSI, pneumonia severity index.

**Table 2 antibiotics-09-00577-t002:** Length of hospital stay in days by CCI category and PSI class among patients with CABP who received ceftriaxone and macrolide on day 1 or 2 of hospitalization.

CCI Score	Overall			Risk Classes I and II			Risk Class III			Risk Class IV			Risk Class V		
	*n*	Mean	SD	*n*	Mean	SD	*n*	Mean	SD	*n*	Mean	SD	*n*	Mean	SD
**0**	7303	5.1	3.0	4060	4.9	2.8	2430	5.3	3.2	796	5.6	3.3	17	6.4	2.2
**1**	9056	5.4	3.3	4288	5.1	3.0	3301	5.5	3.3	1414	6.1	3.9	53	9.3	8.0
**2**	6295	6.0	3.8	1876	5.4	3.2	2462	5.9	3.5	1854	6.5	4.3	103	9.3	7.9
**3+**	10,263	6.8	4.7	1365	6.1	4.1	2907	6.2	3.8	5285	7.0	4.7	706	8.9	7.5
**Total**	32,917	5.88	3.90	11,589	5.18	3.15	11,100	5.75	3.47	9349	6.63	4.42	879	8.94	7.5

CABP, community-acquired bacterial pneumonia; CCI, Charlson comorbidity index; PSI, pneumonia severity index; SD, standard deviation.

## Data Availability

Data not publicly available.

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
