# Peer review of "Hospital Admission Patterns in Adult Patients with Community-Acquired Pneumonia Who Received Ceftriaxone and a Macrolide by Disease Severity across United States Hospitals"

_antibiotics, 2020, doi:10.3390/antibiotics9090577_

Round 1

Reviewer 1 Report

Dear authors,

I have read the manuscript antibiotics-901052  thoroughly. The short report on the use of cephalosporin and a macrolide in elder patients with pneumonia in US hospitals is well structured and written and contains information that maybe of interest to the readers of this magazine. I beleive that the article is also in accordance with the scope of the magazine and I suggest it is published after two typing mistakes are corrected:

Line 82: Patients were included if they were age …

Line 168: …there is an opportunity to improve the …

Kind regards,

Author Response

Reviewer 1

I have read the manuscript antibiotics-901052  thoroughly. The short report on the use of cephalosporin and a macrolide in elder patients with pneumonia in US hospitals is well structured and written and contains information that maybe of interest to the readers of this magazine. I beleive that the article is also in accordance with the scope of the magazine and I suggest it is published after two typing mistakes are corrected:

Line 82: Patients were included if they were age …

  • Recommendation incorporated.

Line 168: …there is an opportunity to improve the …

  • Recommendation incorporated.

Reviewer 2 Report

Dear authors,

Manuscript entitled "Hospital admission patterns in adult patients with community acquired pneumonia" is well drafted and very huge patient population was considered for the study. 

Authors had published similar type of study titled "Hospital Admission Patterns in Adult Patients with Community-Acquired Bacterial Pneumonia Who Received Ceftriaxone and a Macrolide by Pneumonia Severity Index Score” in Open Forum Infectious Diseases. Conclusions of the studies are almost similar in both the papers with no further new observations in the latest manuscript.

As mentioned, further investigations in the clinical studies and admission pattern data could have provided more novelty to the work.

Author Response

Reviewer 2

Dear authors,

Manuscript entitled "Hospital admission patterns in adult patients with community acquired pneumonia" is well drafted and very huge patient population was considered for the study. 

Authors had published similar type of study titled "Hospital Admission Patterns in Adult Patients with Community-Acquired Bacterial Pneumonia Who Received Ceftriaxone and a Macrolide by Pneumonia Severity Index Score” in Open Forum Infectious Diseases. Conclusions of the studies are almost similar in both the papers with no further new observations in the latest manuscript.

As mentioned, further investigations in the clinical studies and admission pattern data could have provided more novelty to the work.

  • Thank you for the feedback. Please note that this study was presented as a poster presentation at the IDweek 2018.  The abstracts from IDweek are published in Open Forum Infectious Diseases https://www.ncbi.nlm.nih.gov/pmc/articles/PMC6252413/.  All the abstracts presented at IDweek 2018 can be found at https://www.ncbi.nlm.nih.gov/pmc/issues/324522/.  
  • To make this clearer, the following was added to the acknowledgements, “This study was presented, in part, as a poster presentation (Abstract # 1462) at IDweek2018™, San Francisco, CA. October 3-7, 2018.”

Reviewer 3 Report

The manuscript of Lodise et al. entitled “Hospital Admission Patterns in Adult Patients with Community-acquired Pneumonia Who Received Ceftriaxone and a Macrolide by Disease Severity Across United States Hospitals” is interesting. This work explains that two-third of CAP have been treated during hospitalizations whereas they could have been treated with a short hospitalization or with outpatient setting. Some minor points have to be improved or added before acceptance.

Minor points

I advise the authors to add a flowchart with included patients, excluded patients, final study population…to make the results easier to read.

I suggest to the authors to add some t-test and pvalue in order to compare length of hospital stay, mortality, readmission between the different classes.

The authors explain line 155 and line 156 that “the lower PSI risk class patients were not being admitted do to underlying comorbidities” because CCI <1. How can we explain these hospitalizations? Are there other factors (such as bacteria resistance)? Or is it an abuse of clinicians?

What about the other countries? Do they have the same observations concerning CAP hospitalizations? Can we generalize the results of this study? Or do these results specifically reflect US practice? Finally, what about the external validity of this study? A very good review published in December 2018 by Peyroni et al gives some answers. I advise the authors to add this reference in their manuscript (PMID 30596308)

Can we conclude that PSI is an unsuitable score to allow hospitalization? Can the authors propose another score according to the criteria of CAP hospitalization? Included patient number is enough to design another score thanks to this cohort. Then new prospective study can be proposed to validate this score and published in a futher paper.

Have the authors an idea of the cost due to abused hospitalizations for CAP in their cohort?

Supplemental Table 4: Risk Class V is written twice instead of Risk Class IV and Risk Class V.

Table 2 title: First word is missing (“length”?)

Author Response

Reviewer 3

The manuscript of Lodise et al. entitled “Hospital Admission Patterns in Adult Patients with Community-acquired Pneumonia Who Received Ceftriaxone and a Macrolide by Disease Severity Across United States Hospitals” is interesting. This work explains that two-third of CAP have been treated during hospitalizations whereas they could have been treated with a short hospitalization or with outpatient setting. Some minor points have to be improved or added before acceptance.

Minor points

I advise the authors to add a flowchart with included patients, excluded patients, final study population…to make the results easier to read.

  • Unfortunately, we are unable to provide any additional granularity on the derivation of the final study population beyond the following, “The database included 284,549 patients with CABP, which represented 132,473 inpatient admissions. During the study period, 32,917 of the 132,473 hospitalized patients with CABP received ceftriaxone and a macrolide.”
  • When the database was constructed, exclusion criteria were applied simultaneously, and we cannot retrace how many patients were eliminated at each step. We do not believe this detracts from the final manuscript as the language in the manuscripts reflects the final number of individuals in the final database. 

I suggest to the authors to add some t-test and pvalue in order to compare length of hospital stay, mortality, readmission between the different classes.

  • Thank you for the recommendation. While reasonable, we believe this request is outside the scope of the study.  The primary goal of this study was to estimate the proportion of hospital CABP admissions that occur among low risk patients (PSI risk classes I–III) where outpatient treatment or a brief inpatient stay is advocated.  We do not believe comparisons of LOS, mortality and readmissions between PSI risk classes is within study scope. The requested analysis would result in the reporting of multiple outcome comparisons across risk groups, which we believe would cloud the study findings.  We are not entirely certain how to interpret the requested analyses as we did not set forth any hypotheses regarding differences in these endpoints across PSI risk classes.  We believe this question is best answered in a different study dedicated to addressing this hypothesis as we anticipated more sophisticated analyses are needed to properly compare outcomes across the different PSI risk classes.  Ultimately, the goal of this brief report was simply to determine if US hospitals are potentially over-admitting patients.  We purposefully summarized the findings as a brief report to keep the manuscript concise and focused. 

The authors explain line 155 and line 156 that “the lower PSI risk class patients were not being admitted do to underlying comorbidities” because CCI <1. How can we explain these hospitalizations? Are there other factors (such as bacteria resistance)? Or is it an abuse of clinicians?

  • Although we could not ascertain the exact reasons for admissions, we anticipate “that inconsistent criteria are being applied for initial decisions about appropriate site of care and there is a potential opportunity to shift site of care for many hospitalized patients with CABP.”
  • However, we acknowledge that the PSI score may have been underestimated in certain CCI<1 patients and other reasons for hospital admission (for example, socioeconomic factors, inability to tolerate oral medications, etc.) could be responsible for the admissions among CCI <1 patients. We revised the discussion (see revised paper) and added to following to the paper to make this clearer, “It is also important to note that the initial severity of a patient’s CAP is not the only predictor of hospital admission and LOS. Community-acquired pneumonia may worsen other comorbidities such as chronic heart failure, COPD and renal failure, which may increase hospitalization time. The influence of this aspect of CAP hospitalization was not evaluated in this study and should be investigated further. We were also not able to fully elucidate the clinical and non-clinical (e.g. socioeconomic) reasons for hospital admission among these patients. We also could not determine if patients would have had similar or better outcomes with an alternative site of care. The intent of this study was simply to highlight the proportion of adult, hospitalized CAP patients that is considered “low risk” and could potentially be managed in the outpatient setting. Given the annual burden of CAP in the US (i.e., 1 million admissions per year) [4], even a conservative application of our findings suggest that there is a tremendous opportunity to improve the efficiency of healthcare delivery for CABP patients by shifting inpatient care to the outpatient setting in appropriate patients.”

What about the other countries? Do they have the same observations concerning CAP hospitalizations? Can we generalize the results of this study? Or do these results specifically reflect US practice? Finally, what about the external validity of this study? A very good review published in December 2018 by Peyroni et al gives some answers. I advise the authors to add this reference in their manuscript (PMID 30596308)

  • Thank you for recommending the paper by Peyroni and colleagues. It is a very good read. 
  • For this paper, we prefer not to compare our US findings to other countries. To do this properly, we would need to apply the same inclusion/exclusion criteria, and this is not possible without the source data. 
  • With regards to external validity, we believe the findings are reflective of practices common throughout the US since the analysis included data from more than 400 US hospitals. We reviewed the paper closely and we prefer to keep the Discussion in its current conservative form.  We believe the most appropriate, albeit conservative, interpretation of the data is that “Consistent with older data [3, 9, 10], the findings suggest that inconsistent criteria may be applied for initial decisions about appropriate site of care and there is a potential opportunity to shift site of care for many hospitalized patients with CABP.  Although the results should be interpreted cautiously due to the nature of the analyses, the findings can potentially serve as the basis for quality assurance initiatives in hospitals seeking to improve the efficiency of healthcare delivery for CABP patients.”  We also believe that, as stated in the conclusions, “Similar to other healthcare database studies, further investigations in the clinical arena are needed.  It will be important to determine if the admission patterns by PSI risk class observed in this study are consistent with future studies of a similar nature that use laboratory data to calculate the PSI score.”  We prefer not to make any additional statements beyond this but defer to the editor.       

Can we conclude that PSI is an unsuitable score to allow hospitalization? Can the authors propose another score according to the criteria of CAP hospitalization? Included patient number is enough to design another score thanks to this cohort. Then new prospective study can be proposed to validate this score and published in a further paper.

  • The reviewer raises a critical point. Unfortunately, our study was not designed to assess the suitability of the PSI scoring system for admission decisions.  Rather, we calculated the PSI score for each patient and simply assessed admission patterns.  The updated CAP guidelines still advocate for the PSI scoring system and recommend it over other disease severity scores, including CURB-65, due to its relatively higher sensitivity and specificity.  
  • Most importantly, we believe the findings from this study indicate that clinicians are not considering the PSI score when making site of care decisions in patients. This is the most critical finding from this study (clinicians do not follow the expert guidelines).  We are hopeful that the revisions to the paper make this point clearer. 
  • We would love to conduct the study proposed by the reviewer. However, we would need access to electronic medical record data to devise a new initial site of care scoring system. At this point, we believe the next step is as follows, ““Similar to other healthcare database studies, further investigations in the clinical arena are needed.  It will be important to determine if the admission patterns by PSI risk class observed in this study are consistent with future studies of a similar nature that use laboratory data to calculate the PSI score.”  We prefer not to make any additional statements but defer to the editor.      

Have the authors an idea of the cost due to abused hospitalizations for CAP in their cohort?

  • We believe the costs associated with abused hospitalizations are substantial given each CAP admission costs ~13,000 USD. However, we prefer not to speculate on the excess CAP admission costs since we do not know the definitive reason why patients were admitted in this study. Rather, we prefer to include the following generic statement in the discussion, “Given the annual burden of CAP in the US (i.e., 1 million admissions per year) [4], even a conservative application of our findings suggest that there is a tremendous opportunity to improve the efficiency of healthcare delivery for CABP patients by shifting inpatient care to the outpatient setting in appropriate patients.”  We believe this captures the essence of the point made by the reviewer.

Supplemental Table 4: Risk Class V is written twice instead of Risk Class IV and Risk Class V.

  • Recommendation incorporated. Thank you. 

Table 2 title: First word is missing (“length”?)

  • Recommendation incorporated. Thank you.